# FEDERATED RECOMMENDATION WITH ADDITIVE PERSONALIZATION

**Zhiwei Li**[1] **Guodong Long**[1] **Tianyi Zhou**[2]

[1] Australian AI Institute, Faculty of Engineering and IT, University of Technology Sydney
[2] Department of Computer Science, University of Maryland, College Park
zhw.li@outlook.com, guodong.long@uts.edu.au, zhou@umiacs.umd.edu

## ABSTRACT

Building recommendation systems via federated learning (FL) is a new emerging challenge for next-generation Internet service. Existing FL models share item embedding across clients while keeping the user embedding private and local on the client side. However, identical item embedding cannot capture users' individual differences in perceiving the same item and may lead to poor personalization. Moreover, dense item embedding in FL results in expensive communication costs and latency. To address these challenges, we propose **Fed**erated **R**ecommendation with **A**dditive **P**ersonalization (**FedRAP**), which learns a global view of items via FL and a personalized view locally on each user. FedRAP encourages a sparse global view to save FL's communication cost and enforces the two views to be complementary via two regularizers. We propose an effective curriculum to learn the local and global views progressively with increasing regularization weights. To produce recommendations for a user, FedRAP adds the two views together to obtain a personalized item embedding. FedRAP achieves the best performance in FL setting on multiple benchmarks. It outperforms recent federated recommendation methods and several ablation study baselines. Our code is available at https://github.com/mtics/FedRAP.

## 1 INTRODUCTION

Recommendation systems have emerged as an important tool and product to allocate new items a user is likely to be interested in and they deeply change daily lives (Deng et al., 2019). These systems typically rely on servers to aggregate user data, digital activities, and preferences in order to train a model to make accurate recommendations (Das et al., 2017; Zhang et al., 2023b). However, uploading users' personal data, which often contains sensitive privacy information, to central servers can expose them to significant privacy and security risks (Chai et al., 2020). Furthermore, recent government regulations on privacy protection (Voigt & Von dem Bussche, 2017) necessitate that user data need to be stored locally on their devices, rather than being uploaded to a global server. As a potential solution to the above problem, federated learning (FL) (McMahan et al., 2017) enforces data localization and trains a globally shared model in a distributed manner, to be specific, by alternating between two fundamental operations, i.e., client-side local model training and server-side aggregation of local models. It achieves great success in some applications such as Google keyboard query suggestions (Hard et al., 2018; Chen et al., 2019). However, the heterogeneity across clients, e.g., non-identical data distributions, can significantly slow down the FL convergence, resulting in client drift or poor global model performance on individual clients. A variety of non-IID FL methods (Zhao et al., 2018; Li et al., 2020; Karimireddy et al., 2020b; Ma et al., 2022; Yan & Long, 2023) is developed to address the challenge by finding a better trade-off between global consensus and local personalization in model training. However, most of them mainly focus on similar-type classification tasks, which naturally share many common features, and few approaches are designed for recommendation systems, which have severer heterogeneity among users, focus more on personalization, but suffer more from local data deficiency.

**Federated Recommendation.** In the quest to foster knowledge exchange among heterogeneous clients without breaching user privacy, scholars are delving into federated recommendation systems. This has given rise to the burgeoning field of Federated Recommendation Systems (FRSs) (Zhang

et al., 2021). FRSs deal with client data originating from a single user, thereby shaping the user's profile (Imran et al., 2023; Yuan et al., 2023b). The user profile and rating data stay locally on the client side while the server is allowed to store candidate items' information. To adhere to FL constraints while preserving user privacy, FRSs need to strike a fine balance between communication costs and model precision to yield optimal recommendations (Tan et al., 2022b). Several recent approaches (Lin et al., 2020; Li et al., 2020; Liang et al., 2021; Yuan et al., 2023a; Zhang et al., 2023c) have emerged to confront these hurdles. Most of them share a partial model design (Singhal et al., 2021; Pillutla et al., 2022) in which the item embedding is globally shared and trained by the existing FL pipeline while the user function/embedding is trained and stays local. However, they ignore the heterogeneity among users in perceiving the same item, i.e., users have different preference to each item and they may focus on different attributes of the item. While PFedRec (Zhang et al., 2023a) considers personalized item embedding, it neglects the knowledge sharing and collaborative filtering across users via global item embedding. Moreover, FL of item embedding requires expensive communication of a dense matrix between clients and a server, especially for users interested in many items.

**Main Contributions.** To bridge the gap described above in FRSs, we propose **Fed**erated **R**ecommendation with **A**dditive **P**ersonalization (**FedRAP**), which balances global knowledge sharing and local personalization by applying an additive model to item embedding and reduces communication cost/latency with sparsity. FedRAP follows horizontal FL assumption (Alamgir et al., 2022) with distinct user embedding and unique user datasets but shared items. Specifically, the main contributions of FedRAP are as follows:

- Unlike previous methods, FedRAP integrates a two-way personalization: personalized and private user embedding, and item additive personalization via summation of a user-specific item embedding matrix $\mathbf{D}^{(i)}$ and a globally shared item embedding matrix $\mathbf{C}$ updated via server aggregations.

- Two regularizers are applied: one encouraging the **sparsity of $\mathbf{C}$ (to reduce the communication cost/overhead)** and the other **enforcing difference** between $\mathbf{C}$ and $\mathbf{D}^{(i)}$ (to be complementary).

- In earlier training, additive personalization may hamper performance due to the time-variance and overlapping between $\mathbf{C}$ and $\mathbf{D}^{(i)}$; to mitigate this, regularization weights are incrementally increased by **a curriculum transitioning from full to additive personalization**.

Therefore, FedRAP is able to utilize locally stored partial ratings to predict user ratings for unrated items by **considering both the global view and the user-specific view of the items.** In experiments on six real-data benchmarks, FedRAP significantly outperforms SOTA FRS approaches.

## 2 RELATED WORK

**Personalized Federated Learning.** To mitigate the effects of heterogeneous and non-IID data, many works have been proposed (Yang et al., 2018; Zhao et al., 2018; Ji et al., 2019; Karimireddy et al., 2020a; Huang et al., 2020; Luo et al., 2021; Li et al., 2021; Wu & Wang, 2021; Hao et al., 2021; Wang et al., 2022; Long et al., 2023; Luo et al., 2023; Ma et al., 2023). In this paper, we focus on works of personalized federated learning (PFL). Unlike the works that aim to learn a global model, PFL seeks to train individualized models for distinct clients (Tan et al., 2022a), often necessitating server-based model aggregation (Arivazhagan et al., 2019; T Dinh et al., 2020; Collins et al., 2021). Several studies (Ammad-Ud-Din et al., 2019; Huang et al., 2020; T Dinh et al., 2020) accomplish personalized federated learning by introducing various regularization terms between local and global models. Meanwhile, some works (Flanagan et al., 2020; Li et al., 2021; Luo et al., 2022; Tan et al., 2023) focus on personalized model learning, with the former promoting the closeness of local models via variance metrics and the latter enhancing this by clustering users into groups and selecting representative users for training, instead of random selection.

**Federated Recommendation Systems.** FRSs are unique in that they typically have a single user per client (Sun et al., 2022), with privacy-protective recommendations garnering significant research attention. Several FRS models leverage stochastic gradient updates with implicit feedback (Ammad-Ud-Din et al., 2019) or explicit feedback (Lin et al., 2020; Liang et al., 2021; Perifanis & Efraimidis, 2022; Luo et al., 2022). FedMF (Chai et al., 2020) introduces the first federated matrix factorization algorithm based on non-negative matrix factorization (Lee & Seung, 2000), albeit with limitations

in privacy and heterogeneity. Meanwhile, PFedRec (Zhang et al., 2023a) offers a bipartite personalization mechanism for personalized recommendations, but neglects shared item information, potentially degrading performance. Although existing works have demonstrated promising results, they typically only consider user-side personalizations, overlooking diverse emphases on the same items by different users. In contrast, FedRAP employs a bipartite personalized algorithm to manage data heterogeneity due to diverse user behaviors, considering both unique and shared item information.

## 3 FEDERATED RECOMMENDATION WITH ADDITIVE PERSONALIZATION

Taking the fact that users' preferences for items are influenced by the users into consideration, in this research, we assume that the ratings on the clients are determined by the users' preferences, which are affected by both the user information and the item information. Thus, the user information is varied by different users, and the item information on each client should be similar, but not the same. Given partial rating data, the goal of FedRAP is to recommend the items users have not visited.

**Notations.** Let $\mathbf{R} = [r_1, r_2, \ldots, r_n]^T \in \{0, 1\}^{n \times m}$ denote that the input rating data with $n$ users and $m$ items, where $r_i \in \{0, 1\}^m$ indicates the $i$-th user's rating data. Since each client only have one user information as stated above, $r_i$ also represents the $i$-th client's ratings. Moreover, we use $\mathbf{D}^{(i)} \in \mathbb{R}^{m \times k}$ to denote the local item embedding on the $i$-th client, and $\mathbf{C} \in \mathbb{R}^{m \times k}$ to denote the global item embedding. In the recommendation setting, one user may only rate partial items. Thus, we introduce $\mathbf{\Omega} = \{(i, j) : \text{the } i\text{-th user has rated the } j\text{-th item}\}$ to be the set of rated entries in $\mathbf{R}$.

### 3.1 PROBLEM FORMULATION

To retain shared item information across users while enabling item personalization, we propose to use $\mathbf{C}$ to learn shared information, and employ $\mathbf{D}^{(i)}$ to capture item information specific to the $i$-th user on the corresponding $i$-th client, where $i = 1, 2, \ldots, n$. FedRAP then uses the summation $\mathbf{C}$ and $\mathbf{D}^{(i)}$ to achieve an additive personalization of items. Considering $\mathbf{R} \in \{0, 1\}^{n \times m}$, we utilize the following formulation for the $i$-th client to map the predicted rating $\hat{r}_{ij}$ into $[0, 1]$:

$$\min_{\mathbf{U}, \mathbf{C}, \mathbf{D}^{(i)}} \sum_{(i,j) \in \mathbf{\Omega}} -(r_{ij} \log \hat{r}_{ij} + (1 - r_{ij}) \log(1 - \hat{r}_{ij})). \tag{1}$$

Here, $\hat{r}_{ij} = 1/(1 + e^{-<\mathbf{u_i},(\mathbf{D}^{(i)}+\mathbf{C})_j>})$ represents the predicted rating of the $i$-th user for the $j$-th item. Eq. (1) minimizes the reconstruction error between the actual rating $r_i$ and predicted ratings $\hat{r}_{ij}$ based on rated entries indexed by $\mathbf{\Omega}$. To ensure the item information learned by $\mathbf{D}^{(i)}$ and $\mathbf{C}$ differs on the $i$-th client, we enforce the differentiation among them with the following equation:

$$\max_{\mathbf{C}, \mathbf{D}^{(i)}} \sum_{i=1}^{n} ||\mathbf{D}^{(i)} - \mathbf{C}||_F^2. \tag{2}$$

Eqs. (1) and (2) describe the personalization model for each client, and aim to maximize the difference between $\mathbf{D}^{(i)}$ and $\mathbf{C}$. We note that in the early stages of learning, the item information learned by $\{\mathbf{D}^{(i)}\}_{i=1}^n$ and $\mathbf{C}$ may be inadequate for recommendation, and additive personalization may thus reduce performance. Thus, considering Eqs. (1), (2), and the aforementioned condition, we propose the following optimization problem $\mathcal{L}$ in FedRAP:

$$\min_{\mathbf{U}, \mathbf{C}, \mathbf{D}^{(i)}} \sum_{i=1}^{n} \left( \sum_{(i,j) \in \mathbf{\Omega}} -(r_{ij} \log \hat{r}_{ij} + (1 - r_{ij}) \log(1 - \hat{r}_{ij})) - \lambda_{(a,v_1)} ||\mathbf{D}^{(i)} - \mathbf{C}||_F^2 \right) + \mu_{(a,v_2)} ||\mathbf{C}||_1, \tag{3}$$

where $a$ is the number of training iterations. We employ the functions $\lambda_{(a,v_1)}$ and $\mu_{(a,v_2)}$ to control the regularization strengths, limiting them to values between 0 and $v_1$ or $v_2$, respectively. Here, $v_1$ and $v_2$ are constants that determine the maximum values of the hyperparameters. By incrementally boosting the regularization weights, FedRAP gradually transitions from full personalization to additive personalization concerning the item information. As there may be some redundant information in $\mathbf{C}$, we employ the $L_1$ norm on $\mathbf{C}$ ($||\mathbf{C}||_1$) to induce $\mathbf{C}$ sparsity, which eliminates the unnecessary information from $\mathbf{C}$ and helps reduce the communication cost between the server and the client. The whole framework of FedRAP is depicted in Fig. 1.

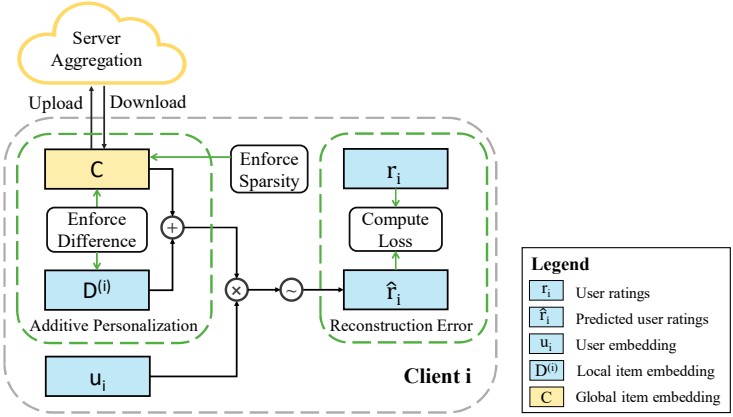

Figure 1: **The framework of FedRAP**. For each user-$i$, FedRAP utilizes its ratings $r_i$ as labels to locally train a user embedding $\mathbf{u}_i$ and a local item embedding $\mathbf{D}^{(i)}$. By adding $\mathbf{D}^{(i)}$ to the global item embedding $\mathbf{C}$ from the server, i.e., **additive personalization**, FedRAP creates a personalized item embedding based on both shared knowledge and personal perspective and thus produces better recommendations. Since clients and server only communicate $\mathbf{C}$, FedRAP enforce its sparsity to reduce the communication cost and encourage its difference to $\mathbf{D}^{(i)}$ so they are complementary.

## 3.2 ALGORITHM

**Algorithm Design.** To address the objective described in Eq. (3), we employ an alternative optimization algorithm to train our model. As shown in Alg. 1, the overall workflow of the algorithm is summarized into several steps as follows. We start by initializing $\mathbf{C}$ at the server and $\mathbf{u}_i$ and $\mathbf{D}^{(i)}$ at their respective clients ($i = 1, 2, \ldots, n$). Each client is assumed to represent a single user. For every round, the server randomly selects a subset of clients, represented by $S_a$, to participate in the training. Clients initially receive the sparse global item embedding $\mathbf{C}$ from the server. They then invoke the function $\mathrm{ClientUpdate}$ to update their parameters accordingly with the learning rate $\eta$. Clients upload the updated $\{\mathbf{C}^{(i)}\}_{i=1}^{n_s}$ to the server for aggregation. The server then employs the aggregated $\mathbf{C}$ in the succeeding training round. Upon completion of the training process, FedRAP outputs the predicted ratings $\hat{\mathbf{R}}$ to guide recommendations. FedRAP upholds user privacy by keeping user-specific latent embeddings on clients and merely uploading the updated sparse global item embedding to the server. This approach notably reduces both privacy risk and communication costs. Furthermore, FedRAP accommodates user behavioral heterogeneity by locally learning user-specific personalized models at clients. As such, even though user data varies due to individual preferences, FedRAP ensures that data within a user adheres to the i.i.d assumption.

**Cost Analysis.** Regarding the time cost, following Algorithm 1, the computational complexity on the $i$-th client for updating $\mathbf{u}_i$, $\mathbf{D}^{(i)}$, and $\mathbf{C}$ requires a cost of $\mathcal{O}(km)$. Once the server receives $\mathbf{C}$ from $n$ clients, it takes $\mathcal{O}(kmn)$ for the server to perform aggregations. Consequently, the total computational complexity of Algorithm 1 at each iteration is $\mathcal{O}(kmn)$. In practice, the algorithm typically converges within 100 iterations. As for the space cost, each client needs to store $\mathbf{u}_i \in \mathbb{R}^k$, $\mathbf{D}^{(i)} \in \mathbb{R}^{(m \times k)}$, and $\mathbf{C} \in \mathbb{R}^{(m \times k)}$, requiring $\mathcal{O}((2nm+n)k+nm)$ space. The server, on the other hand, needs to store the updated and aggregated global item embedding, requiring $\mathcal{O}((nm+m)k)$ space. Thus, FedRAP demands a total of $\mathcal{O}((3nm+n+m)k+nm)$ space.

## 3.3 DIFFERENTIAL PRIVACY

Despite the fact that FedRAP only conveys the global item embedding $\mathbf{C}$ between the server and clients, there still lurks the potential risk of information leakage.

**Property 1 (Revealing local gradient from consecutive C)** *If a server can obtain consecutive $\mathbf{C}^{(i)}$ returned by client $i$, it can extract the local gradient with respect to $\mathbf{C}$ from the latest update of the corresponding client.*

---

**Algorithm 1** Federated Recommendation with Additive Personalization (FedRAP)

---

**Input**: $\mathbf{R}$, $\boldsymbol{\Omega}$, $v_1$, $v_2$, $k$, $\eta$, $t_1$, $t_2$
**Initialize**:$\mathbf{C}$
**Global Procedure**:
 1: **for** each client index $i = 1, 2, \ldots, n$ **in parallel do**
 2:      initialize $\mathbf{u}_i$, $\mathbf{D}^{(i)}$;
 3: **end for**
 4: **for** $a = 1, 2, \ldots, t_1$ **do**
 5:      $S_a \leftarrow$ randomly select $n_s$ from $n$ clients
 6:      **for** each client index $i \in S_a$ **in parallel do**
 7:         download $\mathbf{C}$ from the server;
 8:         $\mathbf{C}^{(i)}, \hat{r}_i \leftarrow \text{ClientUpdate}(\mathbf{u}_i, \mathbf{C}, \mathbf{D}^{(i)})$;
 9:         upload $\mathbf{C}^{(i)}$ to the server;
10:      **end for**
11:      $\mathbf{C} \leftarrow \frac{1}{n_s} \sum_{i=1}^{n_s} \mathbf{C}^{(i)}$;                                ▷ Server Aggregation
12: **end for**
13: **return:** $\hat{\mathbf{R}} = [\hat{r}_1, \hat{r}_2, \ldots, \hat{r}_n]^T$
**ClientUpdate**:
 1: **for** $b = 1, 2, \ldots, t_2$ **do**
 2:      calculate the partial gradients $\nabla_{\mathbf{u}_i}\mathcal{L}$, $\nabla_{\mathbf{C}}\mathcal{L}$ and $\nabla_{\mathbf{D}^{(i)}}\mathcal{L}$ by differentiating Eq. (3);
 3:      update $\mathbf{u}_i \leftarrow \mathbf{u}_i - \eta\nabla_{\mathbf{u}_i}\mathcal{L}$;
 4:      update $\mathbf{C}^{(i)} \leftarrow \mathbf{C} - \eta\nabla_{\mathbf{C}}\mathcal{L}$;
 5:      update $\mathbf{D}^{(i)} \leftarrow \mathbf{D}^{(i)} - \eta\nabla_{\mathbf{D}^{(i)}}\mathcal{L}$;
 6: **end for**
 7: $\hat{r}_i = \sigma(\langle \mathbf{u_i}, \mathbf{D^{(i)}} + \mathbf{C^{(i)}} \rangle)$;
 8: **return:** $\mathbf{C}^{(i)}, \hat{r}_i$

---

**Discussion 1** *If client $i$ participates in two consecutive training rounds (denoted as $a$ and $a+1$), the server obtains $\mathbf{C}^{(i)}$ returned by the client in both rounds, denoted as $\mathbf{C}_{(a)}^{(i)}$ and $\mathbf{C}_{(a+1)}^{(i)}$, respectively. According to Alg. 1, it can be deduced that $\mathbf{C}_{(a+1)}^{(i)} - \mathbf{C}_{(a)}^{(i)} = \eta\nabla\mathbf{C}_{(a+1)}^{(i)}$, where $\eta$ is the learning rate. Since there must exist a constant equal to $\eta$, the server can consequently obtain the local gradient of the objective function with respect to $\mathbf{C}$ for client $i$ during the $(a+1)$-th training round.*

Property 1 shows that a server, if in possession of consecutive $\mathbf{C}^{(i)}$ from a specific client, can derive the local gradient relative to $\mathbf{C}$ from the most recent client update, which becomes feasible to decipher sensitive user data from the client's local gradient (Chai et al., 2020). This further illustrates that a client should not participate in training consecutively twice. Despite our insistence on sparse $\mathbf{C}$ and the inclusion of learning rate $\eta$ in the gradient data collected by the server, the prospect of user data extraction from $\nabla\mathbf{C}_{(a+1)}^{(i)}$ persists.

To fortify user privacy, we employ the widely used $(\epsilon, \delta)$-differential privacy protection technique (Minto et al., 2021) on $\mathbf{C}$, where $\epsilon$ measures the potential information leakage, acting as a privacy budget, while $\delta$ permits a small probability of deviating slightly from the $\epsilon$-differential privacy assurance. To secure $(\epsilon, \delta)$-Differential Privacy, it's imperative for FedRAP to meet certain prerequisites. Initially, following each computation as outlined in Alg. 1, FedRAP secures a new gradient w.r.t. $\mathbf{C}$ on the $i$-th client, denoted by $\nabla\mathbf{C}^{(i)}$, which is limited by a pre-defined threshold $\tau$:

$$\nabla\mathbf{C}^{(i)} \leftarrow \nabla\mathbf{C}^{(i)} \cdot \min\{1, \frac{\tau}{||\nabla\mathbf{C}^{(i)}||_F}\}. \tag{4}$$

Post gradient clipping in Eq. (4), $\nabla\mathbf{C}^{(i)}$ is utilized to update $\mathbf{C}$ on the client $i$. Upon receipt of all updated $\{\mathbf{C}^{(i)}\}_{i=1}^{n_s}$, the server aggregates to derive a new $\mathbf{C}$.

**Property 2 (Sensitivity upper bound of C)** *The sensitivity of $\mathbf{C}$ is upper bounded by $\frac{2\eta\tau}{n_s}$.*

**Discussion 2** *Given two global item embeddings* $\mathbf{C}$ *and* $\mathbf{C}'$, *learned from two datasets that differ only in the data of a single user (denoted as $u$), respectively, we have:*

$$||\mathbf{C} - \mathbf{C}'||_F = ||\frac{\eta}{n_s}(\nabla\mathbf{C}^{(u)} - \nabla\mathbf{C}'^{(u)})||_F \leq \frac{\eta}{n_s}||\nabla\mathbf{C}^{(u)}||_F + \frac{\eta}{n_s}||\nabla\mathbf{C}'^{(u)}||_F \leq \frac{2\eta\tau}{n_s}. \quad (5)$$

Property 2 posits that the sensitivity of $\mathbf{C}$ cannot exceed $\frac{2\eta\tau}{n_s}$. To ensure privacy, we introduce noise derived from the Gaussian distribution $\mathcal{N}(0, e^2)$ to disrupt the gradient, where $e$ is proportionate to the sensitivity of $\mathbf{C}$. As stated in McMahan et al. (2018), applying the Gaussian mechanism $(\frac{2\eta\tau}{n_s}, z)$ during a training round ensures a privacy tuple of $(\frac{2\eta\tau}{n_s}, z)$. Following Abadi et al. (2016), FedRAP leverages the composability and tail bound properties of moment accountants to determine $\epsilon$ for a given budget $\delta$, thus achieving $(\epsilon, \delta)$-differential privacy.

# 4 EXPERIMENTS

## 4.1 DATASETS

A thorough experimental study has been conducted to assess the performance of the introduced FedRAP on six popular recommendation datasets: MovieLens-100K (ML-100K), MovieLens-1M (ML-1M), Amazon-Instant-Video (Video), LastFM-2K (LastFM) (Cantador et al., 2011), Ta Feng Grocery (TaFeng), and QB-article (Yuan et al., 2022). The initial four datasets encompass explicit ratings that range between 1 and 5. As our task is to generate recommendation predictions on data with implicit feedback, any rating higher than 0 in these datasets is assigned 1. TaFeng and QB-article are two implicit feedback datasets, derived from user interaction logs. In each dataset, we include only those users who have rated at least 10 items.

## 4.2 BASELINES

The efficacy of FedRAP is assessed against several cutting-edge methods in both centralized and federated settings for validation.

**NCF (He et al., 2017).** It merges matrix factorization with multi-layer perceptrons (MLP) to represent user-item interactions, necessitating the accumulation of all data on the server. We set up three layers of MLP according to the original paper.

**LightGCN (He et al., 2017).** It is a collaborative filtering model for recommendations that combines user-item interactions without additional complexity from side information or auxiliary tasks.

**FedMF (Chai et al., 2020).** This approach employs matrix factorization in federated contexts to mitigate information leakage through the encryption of the gradients of user and item embeddings.

**FedNCF (Perifanis & Efraimidis, 2022).** It is the federated version of NCF. It updates user embedding locally on each client and synchronizes the item embedding on the server for global updates.

**PFedRec (Zhang et al., 2023a).** It aggregates item embeddings from clients on the server side and leverages them in the evaluation to facilitate bipartite personalized recommendations.

The implementations of NCF, LightGCN, FedMF and PFedRec are publicly available in corresponding papers. Although FedNCF does not provide the code, we have implemented a version based on the code of NCF and the paper of FedNCF. Moreover, we developed a centralized variant of FedRAP, termed **CentRAP**, to showcase the learning performance upper bound for FedRAP within the realm of personalized FRSs.

## 4.3 EXPERIMENTAL SETTING

According to the previous works (He et al., 2017; Zhang et al., 2023a), for each positive sample, we randomly selected 4 negative samples. We performed hyperparameter tuning for all methods, the parameter $v_1$ of our method FedRAP in the range $\{10^i | i = -6, \ldots, 0\}$, and the parameter $v_2$ of FedRAP in the range $\{10^i | i = -3, \ldots, 3\}$. Given that the second and third terms in Eq. 3 gradually come into effect during training, we pragmatically set the function $\lambda_{(a,v_1)} = \tanh(\frac{a}{10}) * v_1$ and

$\mu_{(a,v_2)} = \tanh(\frac{a}{10}) * v_2$, where $a$ is the number of iterations. For PFedRec and FedRAP, the maximum number of server-client communications and client-side training iterations were set to 100 and 10 respectively. For NCF, FedMF and FedNCF, we set the number of training iterations to 200. To be fair, all methods used fixed latent embedding dimensions of 32 and a batch size of 2048. We did not employ any pre-training for any of the methods, nor did we encrypt the gradients of FedMF. Following precedent (He et al., 2017; Zhang et al., 2023a), we used a leave-one-out strategy for dataset split evaluation.

**Evaluation Metrics.** In the experiments, the prediction performances are evaluated by two widely used metrics: *Hit Rate* (HR@K) and *Normalized Discounted Cumulative Gain* (NDCG@K). These criteria has been formally defined in He et al. (2015). In this work, we set $K = 10$, and repeat all experiments by five times. We report the average values and standard deviations of the results.

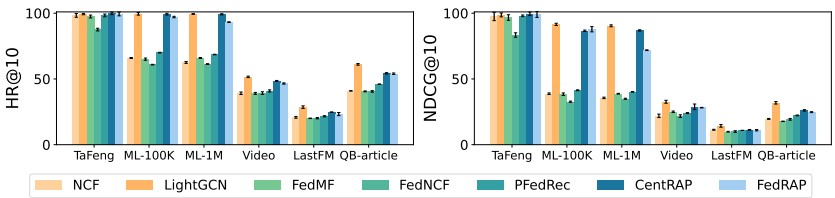

Figure 2: Evaluation metrics (%) on six real-world recommendation datasets. *CentRAP* is a centralized version (upper bound) of *FedRAP*. *FedRAP* outperforms all the FL methods by a large margin.

## 4.4 MAIN RESULTS AND COMPARISONS

Fig. 2 illustrates the experimental results in percentages of all the comparing methods on the six real-world datasets, exhibiting that FedRAP outperforms the others in most scenarios and achieves best among all federated methods. The performance superiority probably comes from the ability of FedRAP on bipartite personalizations of both user information and item information. Moreover, the possible reason why FedRAP performs better than PFedRec is that FedRAP is able to personalize item information while retaining common information of items, thus avoiding information loss. CentRAP exhibits slightly better performance on all datasets compared to FedRAP, demonstrating the performance upper bound of FedRAP on the used datasets. In addition, to investigate the convergence of FedRAP, we compare the evaluations of all methods except CentRAP for each iteration during training on the ML-100K dataset. Because NCF, FedMF and FedNCF are trained of 200 iterations, we record evaluations for these methods every two iterations. Fig. 3 shows that on both HR@10 and NDCG@10, FedRAP achieves the best evaluations in less than 10 iterations and shows a convergence trend within 100 iterations. The experimental results demonstrate the superiority of FedRAP. But since FedRAP is more complex than PFedRec, it needs more iterations to converge.

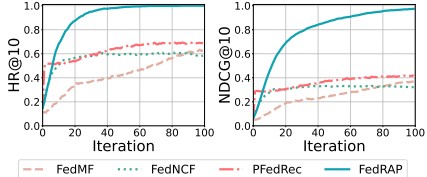

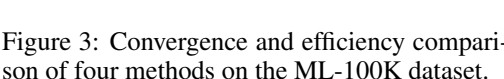

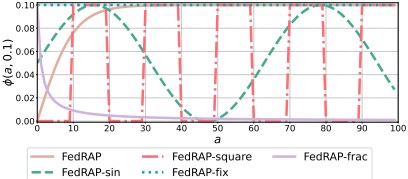

Figure 3: Convergence and efficiency comparison of four methods on the ML-100K dataset.

Figure 4: Different curriculum for $\lambda_{(a,v_1)}$ in Eq. (3) with $v_1 = 0.1$ and $a$ to be the iteration.

## 4.5 ABLATION STUDY

To examine the effects of FedRAP's components, we introduce several variants: FedRAP-C, FedRAP-D, FedRAP-No, FedRAP-L2, FedRAP-fixed, FedRAP-sin, FedRAP-square, and FedRAP-frac, defined as follows: (1) **FedRAP-C**: Uses a common item embedding $\mathbf{C}$ for all users, ignoring personalization; (2) **FedRAP-D**: Creates user-specific item embeddings $\mathbf{D}^{(i)}$, without shared item information; (3) **FedRAP-No**: Removes item sparsity constraint on $\mathbf{C}$, i.e., $||\mathbf{C}||_1$ is omitted; (4)

**FedRAP-L2**: Uses Frobenius Norm, $||\mathbf{C}||_F^2$, instead of $||\mathbf{C}||_1$ for constraint; (5) **FedRAP-fixed**: Fixes $\lambda_{(a,v_1)}$ and $\mu_{(a,v_2)}$ to be two constants $v_1$ and $v_2$, respectively; (6) **FedRAP-sin**: Applies $\lambda_{(a,v_1)} = \sin(\frac{a}{10}) * v_1$ and $\mu_{(a,v_2)} = \sin(\frac{a}{10}) * v_2$; (7) **FedRAP-square**: Alternates the value of $\lambda_{(a,v_1)}$ and $\mu_{(a,v_2)}$ between 0 and $v_1$ or $v_2$ every 10 iterations, respectively; (8) **FedRAP-frac**: Utilizes $\lambda_{(a,v_1)} = \frac{v_1}{a+1}$ and $\mu_{(a,v_2)} = \frac{v_2}{a+1}$. Here, $a$ denotes the number of iterations and $v$ is a constant. FedRAP-C and FedRAP-D validate the basic assumption of user preference influencing ratings. FedRAP-No and FedRAP-L2 assess the impact of sparsity in $\mathbf{C}$ on recommendations. The last four variants explore different weight curricula on FedRAP's performance. We set $v_1$ and $v_2$ to 0.1 for all variants, and Fig. 4 depicts the hyperparameter trends $\lambda_{(a,v_1)}$ of for the final four variants. The trend of $\mu_{(a,v_2)}$ is the same as the corresponding $\lambda_{(a,v_1)}$ of each variant.

**Ablation Study of FedRAP.** To verify the effectiveness of FedRAP, we train all variants as well as FedRAP on the ML-100K dataset. In addition, since PFedRec is similar to FedRAP-D, we also compare it together in this case. As shown in Fig. 5, we plot the trend of HR@10 and NDCG@10 for these six methods over 100 iterations. From Fig. 5, we can see that FedRAP-C performs the worst, followed by FedRAP-D and PFedRec. These indicate the importance of not only personalizing the item information that is user-related, but also keeping the parts of the item information that are non-user-related. In addition, both FedRAP-C and FedRAP-D reach convergence faster than FedRAP. Because FedRAP-C trains a global item embedding $\mathbf{C}$ for all clients, and FedRAP-D does not consider $\mathbf{C}$, while FedRAP is more complex compared to these two algorithms, which explains the slow convergence speed of FedRAP because the model is more complex. The figure also shows that the performance of FedRAP-D and PFedRec is similar, but the curve of FedRAP-D is smoother, probably because both have the same assumptions and similar theories, but we add regularization terms to FedRAP. The results of FedRAP-No, FedRAP-L2 and FedRAP are very close, where the performance of FedRAP is slightly better than the other two as seen in Fig. 5(b). Thus, we choose to use $L_1$ norm to induce $\mathbf{C}$ to become sparse. Also, a sparse $\mathbf{C}$ helps reduce the communication overhead between the server and clients.

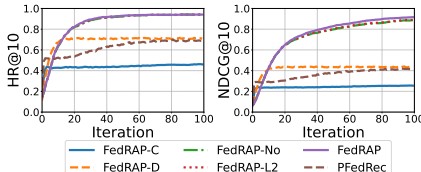

Figure 5: **Ablation study** investigating the effectiveness of FedRAP on the ML-100K dataset.

Figure 6: **Comparison of different curricula** for $\lambda_{(a,v_1)}$ and $\mu_{(a,v_2)}$ in FedRAP on the ML-100K dataset.

**Ablation Study on Curriculum.** In this section, we investigate the suitability of additive personalization in the initial training phase by analyzing the performance of FedRAP and its four weight variations (FedRAP-fixed, FedRAP-sin, FedRAP-square and FedRAP-frac) on HR@10 and NDCG@10 in the validation set of the ML-100K dataset. Fig. 6 shows that FedRAP-fixed, with constant weights, underperforms FedRAP, which adjusts weights from small to large. This confirms that early-stage additive personalization leads to performance degradation, as item embeddings have not yet captured enough useful information. Furthermore, FedRAP-frac, which reduces weights from large to small, performs the worst, while FedRAP performs the worst, reiterating that early-stage additive personalization can decrease performance. Therefore, we choose to use the tanh function.

**Ablation Study on Sparsity of C.** Since FedRAP only exchanges $\mathbf{C}$ between server and clients, retaining user-related information on the client side, we further explore the impact of sparsity constraints on $\mathbf{C}$ in FedRAP by comparing the data distribution of $\mathbf{C}$ and $\{\mathbf{D}^{(i)}\}_{i=1}^n$ learned in the final iteration when training both FedRAP and FedRAP-L2 on the ML-100K dataset. We select three local item embeddings for comparison: $\mathbf{D}^{(43)}$, $\mathbf{D}^{(586)}$, and $\mathbf{D}^{(786)}$, along with the global item embedding $\mathbf{C}$. For visualization, we set latent embedding dimensions to 16 and choose 16 items. Fig. 7 reveals that FedRAP's global item embedding $\mathbf{C}$ is sparser than that of FedRAP-L2. We also notice that 4.19% of FedRAP's $\mathbf{C}$ parameters exceed 1e-1 and 67.36% exceed 1e-2, while for FedRAP-L2,

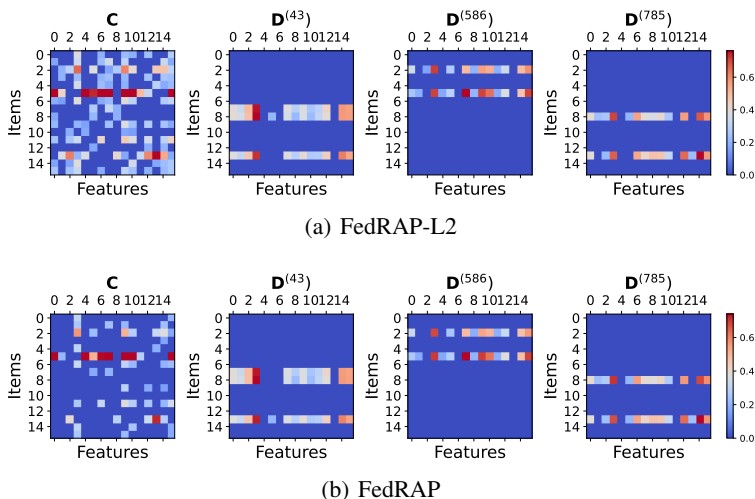

Figure 7: Sparse global matrix $\mathbf{C}$ and row-dense personalized matrix $\mathbf{D}^{(i)}$ for item embedding.

these values are 8.07% and 88.94%, respectively. This suggests that FedRAP's sparsity constraint reduces communication overhead. Additionally, both methods learn similar $\mathbf{D}^{(i)}$ (i = 46, 586, 785), with notable differences between distinct $\mathbf{D}^{(i)}$. This supports our hypothesis that user preferences influence item ratings, leading to data heterogeneity in federated settings and validating FedRAP's ability to learn personalized information for each user.

**FedRAP with Differential Privacy.** To protect user privacy, we introduce noise from the Gaussian distribution to the gradient w.r.t. $\mathbf{C}$ on clients and compare the performance of FedRAP with and without noise. In this work, we set $\tau = 0.1$, and fix the noise variance $z = 1$ for Gaussian Mechanism following the previous work (McMahan et al., 2018). We use the Opacus libaray (Yousefpour et al., 2021) to implement the differential privacy. Table 1 presents results for these approaches on six datasets, showing reduced FedRAP performance after noise addition. However, as Fig 2 indicates, it still outperforms benchmark methods. Moreover, since FedRAP only requires the transmission of $\mathbf{C}$ between the server and clients, adding noise solely to $\mathbf{C}$ is sufficient for ensuring privacy protection. This results in a mitigated performance decline due to differential privacy compared to adding noise to all item embeddings.

Table 1: FedRAP vs. FedRAP-noise (FedRAP with injected noise for **differential privacy**)

| Dataset | TaFeng | | ML-100K | | ML-1M | | Video | | LastFM | | QB-article | |
|---|---|---|---|---|---|---|---|---|---|---|---|---|
| Metrics | HR@10 | NDCG@10 | HR@10 | NDCG@10 | HR@10 | NDCG@10 | HR@10 | NDCG@10 | HR@10 | NDCG@10 | HR@10 | NDCG@10 |
| FedRAP | 0.9943 | 0.9911 | 0.9709 | 0.8781 | 0.9324 | 0.7187 | 0.4653 | 0.2817 | 0.2329 | 0.1099 | 0.5398 | 0.2598 |
| FedRAP-noise | 0.9536 | 0.9365 | 0.9364 | 0.8015 | 0.9180 | 0.7035 | 0.4191 | 0.2645 | 0.2268 | 0.1089 | 0.5020 | 0.2475 |
| Degradation | 4.09%↓ | 5.51%↓ | 3.55%↓ | 8.72%↓ | 1.54%↓ | 2.11%↓ | 9.93%↓ | 6.11%↓ | 2.62%↓ | 0.91%↓ | 7.00%↓ | 4.73%↓ |

## 5 CONCLUSIONS

In this paper, based on the assumption that user's ratings of items are influenced by user preferences, we propose a method named FedRAP to make bipartite personalized federated recommendations, i.e., the personalization of user information and the additive personalization of item information. We achieves a curriculum from full personalization to additive personalization by gradually increasing the regularization weights to mitigate the performance degradation caused by using the additive personalization at the early stages. In addition, a sparsity constraint on the global item embedding to remove useless information for recommendation, which also helps reduce the communication cost. Since the client only uploads the updated global item embedding to the server in each iteration, thus FedRAP avoids the leakage of user information. We demonstrate the effectiveness of our model through comparative experiments on six widely-used real-world recommendation datasets, and numerous ablation studies. Also, due to the simplicity of FedRAP, it would be interesting to explore its applications in other federated scenarios.

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

## A   THEORETICAL ANALYSIS

In this section we give further details on the theoretical results presented in Section 3.4. We start by giving the exact conditions needed for Theorem 1 to hold.

### A.1   FULL DISCUSSION OF THEOREM 1

To delineate Theorem 1, we first re-define the formula for updating $\mathbf{C}^{(i)}$ at the i-th client based on Alg. 1:

$$\mathbf{C}_{(a+1)}^{(i)} = \mathbf{C}_{(a)} - \eta \nabla \mathbf{C}_{(a+1)}^{(i)}, \tag{6}$$

where $\eta$ is the learning rate, and $\nabla \mathbf{C}^{(i)}$ stands for the gradients of Eq. (3) w.r.t. $\mathbf{C}$ on the client $i$ in the $a + 1$-th iteration. When the server receipts all the updated $\{\mathbf{C}_{(a+1)}^{(i)}\}_{i=1}^{n_a}$, it then perform the aggregation to get $\mathbf{C}_{(a+1)}$. Thus, if the client $i$ participate the training in two consecutive rounds, we can get the difference between $\mathbf{C}_{(a+1)}^{(i)}$ and $\mathbf{C}_{(a)}^{(i)}$ by the following fomula:

$$\mathbf{C}_{(a+1)}^{(i)} - \mathbf{C}_{(a)}^{(i)} = \eta \nabla \mathbf{C}_{(a+1)}^{(i)}. \tag{7}$$

Since there exists a constant equal to the pre-defined learning rate $\eta$, an attacker can easily obtain the gradient $\nabla \mathbf{C}_{(a+1)}^{(i)}$ of client $i$ in the $a$-th round from the server. According to Chai et al. (2020), if the client $i$ participates in a sufficient number of training rounds consecutively, the $\mathbf{C}^{(i)}$ obtained by the server would leak information from that client's local data. This emphasizes that a client should avoid participating in training consecutively.

## A.2 OPTIMIZATION ANALYSIS

While the optimization of the objective function is not jointly convex in all variables, it exhibits convexity with respect to each individual variable when the others are held fixed. Consequently, we can pursue an alternate optimization strategy for each of the variables within the objective function in Eq. (3). Denoting $\mathbf{x}_1 = (\mathbf{D}^{(i)} + \mathbf{C})_j$, $\mathbf{x}_2 = 1 + exp(-\mathbf{u}_i^T \mathbf{x}_1)$ and $\mathbf{x}_3 = 1 + exp(-\mathbf{x}_1^T \mathbf{u}_i)$, we can develop an alternating algorithm to update the parameters $\mathbf{U}, \mathbf{C}, \{\mathbf{D}^i\}_{i=1}^n$.

### A.2.1 UPDATE $\mathbf{U}$

Since the updating happens on each client, when $\mathbf{C}$ and $\mathbf{D}^i$ are fixed, the problem w.r.t. $\mathbf{u}_i$ becomes:

$$\min_{\mathbf{u}_i} \sum_{(i,j)\in\mathbf{\Omega}} -(r_{ij} \log 1/(1+e^{-<\mathbf{u_i},(\mathbf{D^{(i)}}+\mathbf{C})_j>}) + (1-r_{ij}) \log(1-1/(1+e^{-<\mathbf{u_i},(\mathbf{D^{(i)}}+\mathbf{C})_j>}))). \quad (8)$$

Assume that $\mathbf{x}_1 = (\mathbf{D}^{(i)} + \mathbf{C})_j$ and $\mathbf{x}_2 = 1 + exp(-\mathbf{u}_i^T x_1)$, we have the partial derivative of $\mathcal{L}(\mathbf{u}_i)$ as following:

$$\nabla_{\mathbf{u}_i}\mathcal{L} = \sum_{(i,j)\in\mathbf{\Omega}} -((e^{(-\mathbf{u}_i^T \mathbf{x_1})} \cdot (1 - r_{ij}))/(\mathbf{x}_2^2 \cdot (1 - 1/\mathbf{x}_2)) \cdot \mathbf{x}_1 + (r_{ij} \cdot e^{(-\mathbf{u}_i^T \mathbf{x_1})})/\mathbf{x}_2 \cdot \mathbf{x}_1). \quad (9)$$

We then update $\mathbf{u}_i$ on each client by gradient descent:

$$\mathbf{u}_i \leftarrow \mathbf{u}_i - \eta\nabla_{\mathbf{u}_i}\mathcal{L}. \quad (10)$$

### A.2.2 UPDATE $\mathbf{C}$

On each client, when fixing $\mathbf{u}_i$ and $\mathbf{D}^{(i)}$, the problem becomes:

$$\min_{\mathbf{C}} \sum_{(i,j)\in\mathbf{\Omega}} -(r_{ij} \log 1/(1 + e^{-<\mathbf{u_i},(\mathbf{D^{(i)}}+\mathbf{C})_j>}) + (1 - r_{ij}) \log(1 - 1/(1 + e^{-<\mathbf{u_i},(\mathbf{D^{(i)}}+\mathbf{C})_j>})))$$
$$+ \sum_{(i,j)\in\mathbf{\Omega}} \lambda_{(a,v_1)}||\mathbf{D}^{(i)} - \mathbf{C}||_F^2 + \mu_{(a,v_2)}||\mathbf{C}||_1. \quad (11)$$

We first calculate the first two terms in Eq. equation 11:

$$\nabla_{\mathbf{C}}\mathcal{L} = \sum_{(i,j)\in\mathbf{\Omega}} -(\frac{(\mathbf{x}_3 - 1) \cdot (1 - r_{ij})}{(\mathbf{x}_3^2 \cdot (1 - \frac{1}{\mathbf{x}_3}))} \cdot \mathbf{u}_i + r_{ij} \cdot (\mathbf{x}_3 - 1)) \cdot \mathbf{u}_i + 2\lambda_{(a,v_1)}(\mathbf{C} - \mathbf{D}^{(i)}). \quad (12)$$

As for the constraint $||\mathbf{C}||_1$, we need to use the soft-thresholding (Donoho, 1995) to update $\mathbf{C}$:

$$\mathbf{C} \leftarrow \mathcal{S}_\theta(\mathbf{C} - \eta\nabla_{\mathbf{C}}\mathcal{L}), \quad (13)$$

where $\theta$ is a pre-defined thresholding value, $\mathcal{S}_\theta$ is the shrinkage operator, and it is defined as $\mathcal{S}_\theta(v) = \max(v - \theta, 0) - \max(-v - \theta, 0)$.

### A.2.3 UPDATE $\mathbf{D}^{(i)}$

When the others fixed, the problem w.r.t. $\mathbf{D}^{(i)}$ becomes:

$$\min_{\mathbf{C}} \sum_{(i,j)\in\mathbf{\Omega}} -(r_{ij} \log 1/(1 + e^{-<\mathbf{u_i},(\mathbf{D^{(i)}}+\mathbf{C})_j>}) + (1 - r_{ij}) \log(1 - 1/(1 + e^{-<\mathbf{u_i},(\mathbf{D^{(i)}}+\mathbf{C})_j>})))$$
$$+ \sum_{(i,j)\in\mathbf{\Omega}} \lambda_{(a,v_1)}||\mathbf{D}^{(i)} - \mathbf{C}||_F^2. \quad (14)$$

Each client then obtains the partial derivative of $\mathcal{L}(\mathbf{D}^{(i)})$ as following:

$$\nabla_{\mathbf{D}^{(i)}}\mathcal{L} = \sum_{(i,j)\in\mathbf{\Omega}} -(\frac{(\mathbf{x}_3 - 1) \cdot (1 - r_{ij})}{(\mathbf{x}_3^2 \cdot (1 - \frac{1}{\mathbf{x}_3}))} \cdot \mathbf{u}_i + r_{ij} \cdot (\mathbf{x}_3 - 1)) \cdot \mathbf{u}_i + 2\lambda_{(a,v_1)}(\mathbf{D}^{(i)} - \mathbf{C}). \quad (15)$$

We can update $\mathbf{D}^{(i)}$ on the client $i$ by the following equation:

$$\mathbf{D}^{(i)} \leftarrow \mathbf{D}^{(i)} - \eta\nabla_{\mathbf{D}^{(i)}}\mathcal{L}. \quad (16)$$

# B   MORE EXPERIMENT DETAILS

In this section, we present some additional details and results of our experiments. To be specific, all models were implemented using PyTorch (Paszke et al., 2019), and experiments were conducted on a machine equipped with a 2.5GHz 14-Core Intel Core i9-12900H processor, a RTX 3070 Ti Laptop GPU, and 64GB of memory.

Table 2: The statistic information of the datasets used in the research. #Ratings is the number of the observed ratings; #Users is the number of users; #Items is the number of items; Sparsity is percentage of #Ratings in (#Users × #Items).

| Datasets | #Ratings | #Users | #Items | Sparsity |
|---|---|---|---|---|
| TaFeng | 100,000 | 120 | 32,266 | 78.88% |
| ML-100k | 100,000 | 943 | 1,682 | 93.70% |
| ML-1M | 1,000,209 | 6,040 | 3,706 | 95.53% |
| Video | 23,181 | 1,372 | 7,957 | 99.79% |
| LastFM | 92,780 | 1,874 | 17,612 | 99.72% |
| QB-article | 266,356 | 24,516 | 7,455 | 99.81% |

## B.1   DATASETS

All the six popular recommendation datasets: MovieLens-100K (ML-100K)[1], MovieLens-1M (ML-1M)[1], Amazon-Instant-Video (Video)[2], LastFM-2K (LastFM)[3], Ta Feng Grocery (TaFeng)[4], and QB-article[5] are public. Table 2 presents the statistical information of the six datasets used in this study. All datasets show a high level of sparsity, with the percentage of observed ratings to the total possible ratings (users times items) all above 90%.

## B.2   COMPRISON ANALYSIS

Tables 3 and 4 present experimental results comparing various centralized and federated methods across six real-world datasets, with the measurements focused on HR@10 and NDCG@10, respectively. In Table 3, it's evident that FedRAP consistently outperforms other federated methods (FedMF, FedNCF, and PFedRec) in terms of HR@10 across all datasets. Remarkably, FedRAP also surpasses the centralized method NCF and approaches the performance of CentRAP, a centralized variant of FedRAP that inherently leverages more data than FedRAP. Table 4 presents a similar trend in terms of NDCG@10. Again, FedRAP demonstrates superior performance among federated methods, and closely competes with the centralized methods, even surpassing CentRAP in some datasets. These results substantiate the effectiveness of FedRAP in balancing personalized item recommendation with privacy preservation in a federated setting. It also highlights FedRAP's capability to handle heterogeneous data across clients while still achieving competitive performance with centralized methods that have full access to all user data.

Table 3: Experimental results on HR@10 shown in percentages on six real-world datasets. **Cent** and **Fed** represent centralized and federated methods, respectively. The best results are highlighted in boldface.

| | Dataset | TaFeng | ML-100K | ML-1M | Video | LastFM | QB-article |
|---|---|---|---|---|---|---|---|
| Cent | NCF | $98.33 \pm 1.42$ | $65.94 \pm 0.38$ | $62.52 \pm 0.70$ | $39.11 \pm 0.95$ | $20.81 \pm 0.69$ | $40.94 \pm 0.16$ |
| | LightGCN | $99.33 \pm 0.51$ | $99.54 \pm 0.99$ | $99.43 \pm 0.31$ | $51.60 \pm 0.54$ | $28.56 \pm 0.97$ | $61.09 \pm 0.63$ |
| | CentRAP | $99.87 \pm 0.97$ | $99.30 \pm 0.64$ | $99.17 \pm 0.43$ | $48.54 \pm 0.32$ | $24.76 \pm 0.12$ | $54.28 \pm 0.51$ |
| Fed | FedMF | $97.50 \pm 1.11$ | $65.05 \pm 0.84$ | $65.95 \pm 0.37$ | $39.00 \pm 0.76$ | $20.04 \pm 0.20$ | $40.54 \pm 0.35$ |
| | FedNCF | $87.54 \pm 1.01$ | $60.89 \pm 0.24$ | $61.31 \pm 0.32$ | $39.25 \pm 1.01$ | $20.14 \pm 0.49$ | $40.52 \pm 0.57$ |
| | PFedRec | $98.33 \pm 1.05$ | $70.03 \pm 0.27$ | $68.59 \pm 0.21$ | $40.87 \pm 0.82$ | $21.61 \pm 0.45$ | $46.00 \pm 0.22$ |
| | FedRAP | $\mathbf{99.43 \pm 1.44}$ | $\mathbf{97.09 \pm 0.56}$ | $\mathbf{93.24 \pm 0.26}$ | $\mathbf{46.53 \pm 0.48}$ | $\mathbf{23.29 \pm 1.19}$ | $\mathbf{53.98 \pm 0.57}$ |

---

[1] https://grouplens.org/datasets/movielens/

[2] http://jmcauley.ucsd.edu/data/amazon/

[3] https://grouplens.org/datasets/hetrec-2011/

[4] https://github.com/RecoHut-Datasets/tafeng

[5] https://github.com/yuangh-x/2022-NIPS-Tenrec

Table 4: Experimental results on NDCG@10 shown in percentages on six real-world datasets. **Cent** and **Fed** represent centralized and federated methods, respectively. The best results are highlighted in boldface.

|  | Dataset | TaFeng | ML-100K | ML-1M | Video | LastFM-2K | QB-article |
|---|---|---|---|---|---|---|---|
| Cent | NCF | $97.55 \pm 3.20$ | $38.76 \pm 0.61$ | $35.60 \pm 0.56$ | $21.94 \pm 1.33$ | $11.36 \pm 0.34$ | $19.61 \pm 0.37$ |
|  | LightGCN | $98.7 \pm 1.47$ | $91.52 \pm 0.86$ | $90.43 \pm 0.72$ | $32.64 \pm 1.13$ | $14.32 \pm 1.01$ | $31.85 \pm 0.96$ |
|  | CentRAP | $99.54 \pm 1.35$ | $86.65 \pm 0.51$ | $86.89 \pm 0.46$ | $28.83 \pm 2.10$ | $11.30 \pm 0.26$ | $26.09 \pm 0.54$ |
| Fed | FedMF | $96.72 \pm 2.13$ | $38.40 \pm 0.97$ | $38.77 \pm 0.29$ | $25.03 \pm 0.57$ | $9.87 \pm 0.25$ | $17.76 \pm 0.18$ |
|  | FedNCF | $83.42 \pm 1.71$ | $32.68 \pm 0.47$ | $34.94 \pm 0.42$ | $21.72 \pm 1.02$ | $9.99 \pm 0.64$ | $19.31 \pm 0.67$ |
|  | PFedRec | $98.03 \pm 0.62$ | $41.38 \pm 0.33$ | $40.19 \pm 0.21$ | $24.04 \pm 0.36$ | $10.88 \pm 0.10$ | $22.28 \pm 0.28$ |
|  | FedRAP | $\mathbf{99.11 \pm 2.31}$ | $\mathbf{87.81 \pm 2.00}$ | $\mathbf{71.87 \pm 0.28}$ | $\mathbf{28.17 \pm 0.22}$ | $\mathbf{10.99 \pm 0.51}$ | $\mathbf{24.75 \pm 0.36}$ |

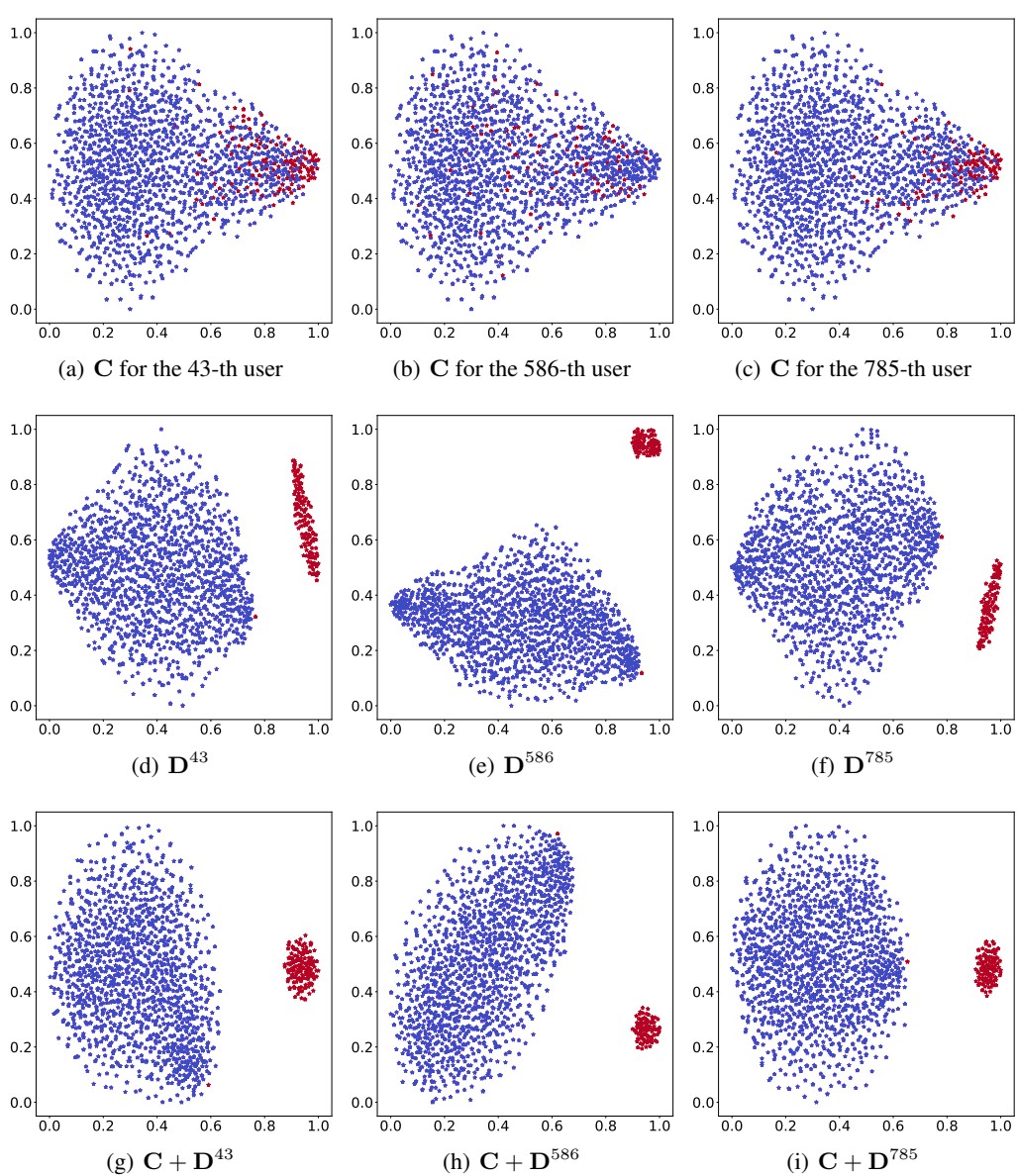

(a) **C** for the 43-th user     (b) **C** for the 586-th user     (c) **C** for the 785-th user

(d) $\mathbf{D}^{43}$       (e) $\mathbf{D}^{586}$       (f) $\mathbf{D}^{785}$

(g) $\mathbf{C} + \mathbf{D}^{43}$       (h) $\mathbf{C} + \mathbf{D}^{586}$       (i) $\mathbf{C} + \mathbf{D}^{785}$

Figure 8: t-SNE visualization of item embeddings learnd by FedRAP.

## B.3 ABLATION STUDY ON ITEM PERSONALIZATION

In this section, we visually demonstrate the benefits of FedRAP for personalizing item information. FedRAP not only personalizes user information, but also implements the additive personalization to item information, thus it achieves the bipartite personalization to perform user-specific recommendations. In this section, we investigate the roles of the global item embedding $\mathbf{C}$ and the local item embeddings $\{\mathbf{D}^{(i)}\}_{i=1}^{n}$ in FedRAP. To visualize them, we select three local item embeddings $\mathbf{D}^{(43)}$, $\mathbf{D}^{(586)}$, $\mathbf{D}^{(785)}$, and the global item embedding $\mathbf{C}$ from the models learned on the MovieLens-100K dataset according to Sec. 4.3. Since in this paper, we mainly focus on the implicit feedback recommendation, i.e., each item is either rated or unrated by the users, we map these item embeddings into a 2-D space through t-SNE (Van der Maaten & Hinton, 2008), and the normalized results are shown in Fig. 8, where purple represents items that have not been rated by the user, and red represents items that have been rated by the user. From Figs. 8 (a)-(c), we can see that the items rated and unrated by users are mixed together in $\mathbf{C}$, which depicts that $\mathbf{C}$ only keeps information about items shared among users. And Figs. 8 (d)-(f) show that the items learned by $\mathbf{D}^{(i)}$ can by obviously divided into two clusters for the $i$-th user by FedRAP. This demonstrates that $\mathbf{D}^{(i)}$ can only learn information about items related to the $i$-th user, such as the user's preferences for items. Morover, we add $\mathbf{C}$ and $\mathbf{D}^{(i)}$ to show the additive personalization of items for the $i$-th user, which is shown in Figs. 8 (g)-(i). These three figures again demonstrate the ability of FedRAP to effectively personalize item information. This case supports the additive personalization of item information to help make user-specific recommendations.

Table 5: Comparison of Performance Degradation with and without Differential Privacy on HR@10.

|  | w/ DP | w/o DP | Degradation |
|---|---|---|---|
| FedMF | 0.6505 | 0.6257 | 3.82% |
| FedNCF | 0.6089 | 0.4348 | 28.60% |
| PFedRec | 0.7003 | 0.6352 | 9.30% |
| FedRAP | 0.9709 | 0.9364 | 3.55% |

Table 6: Comparison of Performance Degradation with and without Differential Privacy on NDCG@10.

|  | w/ DP | w/o DP | Degradation |
|---|---|---|---|
| FedMF | 0.3840 | 0.3451 | 10.13% |
| FedNCF | 0.3268 | 0.2386 | 27.00% |
| PFedRec | 0.4138 | 0.3549 | 14.23% |
| FedRAP | 0.8781 | 0.8015 | 8.72% |

## B.4 ABLATION STUDY ON DIFFERENTIAL PRIVACY

We further evaluated DP versions of the used federated methods and compared them with the original methods in Table 5 and Table 6, which shows FedRAP with DP can still preserve the performance advantage of FedRAP and meanwhile achieve $(\epsilon, \delta)$-DP.

## B.5 ABLATION STUDY ON FACTOR EFFECTS

To examine factors affecting FedRAP's performance, we designed ML-1M dataset variations with different sparsity levels by selecting specific numbers of users or items. Table 7 displays FedRAP's performance on these datasets, and shows that as sparsity increases, performance declines more significantly. Additionally, with similar sparsity, increasing user count leads to slightly reduced performance. Datasets with a fixed number of items consistently underperform compared to those with equal user counts. This table indicates that FedRAP's effectiveness is influenced by dataset sparsity and the counts of users and items.

Table 7: Sparsity of various subsets in ML-1M and FedRAP's fine-grained performance (HR@10 and NDCG@10).

| Subset | Sparsity | HR@10 | NDCG@10 |
|---|---|---|---|
|  | 81.33% | 1.0000 | 0.9446 |
| w/ 500 users | 95.49% | 1.0000 | 0.8227 |
|  | 98.62% | 0.9020 | 0.5686 |
|  | 85.87% | 1.0000 | 0.9064 |
| w/ 1000 users | 97.52% | 0.9990 | 0.7118 |
|  | 98.97% | 0.8260 | 0.5207 |
|  | 87.62% | 0.9851 | 0.8931 |
| w/ 1000 items | 94.86% | 0.9398 | 0.7759 |
|  | 96.85% | 0.9186 | 0.7003 |

## C    DISCUSSION ON LIMIATION

While FedRAP has shown excellent performance in federated recommendation systems through techniques like additive personalization and variable weights, one limitation is that the current Fe-dRAP algorithm requires storing the complete item embedding matrix on each user's device. This could demand significant storage space, especially if the total number of items is large. A simple approach to alleviate this limitation is to only store the embeddings of items that each user has inter-acted with. However, this modification may require extra attention to the cold-start problem of new items for each user. Our future work will aim to address this limitation.

