# OpenReview forum: "Federated Recommendation with Additive Personalization"
_ICLR.cc/2024/Conference — ICLR 2024 poster_

### Official Review · Reviewer_QwLj · 2023-10-30

**Soundness:** 3 good
**Presentation:** 3 good
**Contribution:** 3 good
**Rating:** 6
**Confidence:** 4

**Summary:**

To overcome the challenge of personalization, the authors introduce the Federated Recommendation with Additive Personalization (FedRAP). This method combines global item insights from FL with local, user-specific perspectives. FedRAP ensures efficient communication by promoting sparser global embeddings and uses two regularizers to balance global and local views. The primary contributions include the introduction of the dual personalization approach, the use of two distinct regularizers.

**Strengths:**

1.	The authors have adeptly addressed communication challenges in Federated Learning by advocating for a sparse global view. This practical strategy can potentially alleviate communication bottlenecks.
2.	The emphasis on federated recommendations is commendable, since data privacy is a priority.

**Weaknesses:**

Clarity and Consistency in Writing: The manuscript requires clarity and consistency checks.
a. In Figure 1's title, "five dataset" is mentioned, whereas at the end of Section 1, it states “four dataset”.
b. There seems to be a mismatch in presentation; the results from Section 4.4 on Page 7 are introduced as early as Page 2 in the introduction. Is there a specific reason behind placing the experimental results in the introductory section?

Explicit Statement of Motivation and Research Question: The introductory section should more explicitly state the motivation and the central research question. While Paragraph 3 introduces federated recommendation, and Paragraph 4 delineates the main contribution, there's a lack of clarity on the necessity of addressing heterogeneity and personalization in Federated Recommendation (FedRec). The central research questions around the importance of personalization in FedRec and the inadequacies of existing methods demand elaboration.

Differentiating Heterogeneity from Personalization: The distinction between heterogeneity and personalization needs to be clearer. The manuscript suggests two challenges: heterogeneity, which seems to pertain to varying client interests, and personalization. Given their similarities in this context, why are they presented as two separate challenges?

Organizational Refinements: There's a need for reorganizing the content, especially in Section 3.1 (Problem Formulation). Here, the authors discuss the loss function and the proposed method's entire framework. A more suitable structure might involve mathematically introducing the problem in the Problem Formulation and discussing the method's framework in a separate section.

Clarification on 'Reconstruction Error': Figure 2 illustrates the "reconstruction error". As per my understanding, it seems to measure the discrepancy between actual user ratings and predicted ones. The term "reconstruction error" suggests that something undergoes reconstruction. Can the authors clarify this?

Questioning Novelty: The paper's primary innovation appears to be the maintenance of a user-specific item embedding matrix, D(i), for every user, i. Given that local interactions are usually sparse, which motivates federated training, the uniqueness of this approach is incremental. The two regularizers are essentially the normalization of two embedding tables. While differential privacy is prevalent in federated learning, its application here doesn't seem groundbreaking.

Comparative Analysis with Relevant Works: The manuscript cites PerFedRec [1] in the related works but doesn't present a comparative analysis in the experiments. It would be insightful for the authors to contrast their work with PerFedRec. Moreover, considering the work on PerFedRec++ [2] could be beneficial, even if it's a recent arXiv publication.

References:
[1] Luo, S., Xiao, Y., & Song, L. (2022, October). Personalized federated recommendation via joint representation learning, user clustering, and model adaptation. In Proceedings of the 31st ACM International Conference on Information & Knowledge Management (pp. 4289-4293).
[2] Luo, S., Xiao, Y., Zhang, X., Liu, Y., Ding, W., & Song, L. (2023). PerFedRec++: Enhancing Personalized Federated Recommendation with Self-Supervised Pre-Training. arXiv preprint arXiv:2305.06622.

**Questions:**

1.	Why is the term "Reconstruction error" used in Figure 2? What exactly is being 'reconstructed' in the given context?
2.	Could the authors delineate the difference between 'heterogeneity' and 'personalization'? As of now, the distinction is not abundantly clear, and the terms seem somewhat interchangeable.
3.	The paper emphasizes the sparse embedding's ability to reduce communication costs, a known limitation of current methods. Why is this specific aspect not experimentally validated or discussed in the results section? Can the authors provide insights or potential experimental outcomes supporting this claim?
4.	Given that "PerFedRec" is cited in the related work and seems to be a relevant approach in this domain, why is it omitted from the experimental comparisons? A comparative analysis could provide a clearer perspective on the proposed method's advantages or potential differences.

---

> ### Author Response · Authors · 2023-11-13
> **Rebuttal by Authors**
>
> Thank you for your time and comments! We address your questions and concerns in the following.
>
> ## W1 (Clarity and Consistency in Writing)
> We reported experimental results on six datasets in the paper and we will make the text consistent. We will move the results to Section 4.4 on Page 7.
>
> ## W2 (Statement of Motivation)
>
> In federated recommendation systems, clients or users have different preference to each item and they may focus on different attributes or charateristics of the item, resulting in data heterogeneity, i.e., the items and their ratings are different across clients. Hence, a global model cannot accurately predict the preference of every user, while a personalized model per user can better address the heterogeneity problem.
>
> Most existing personalized federated recommendation systems focus solely on personalizing user embedding or score function, overlooking the potential impact of user preferences on different attributes of items. While PFedRec considers personalized item embedding, it neglects the knowledge sharing and collaborative filtering across users via global item embedding. To bridge the gap, in this work, we introduce a novel additive personalization that takes both the personalization of item embedding and sparse global collaboration in to account, leading to FedRAP that achieves significant improvement on the federated recommendation benchmarks.
>
> We will improve the presentation of our motivation and strengthen the connection between the heterogeneity and the personalization proposed.
>
> ## W3 & Q2 (Differentiating Heterogeneity from Personalization)
>
> Heterogeneity of users in federated recommendation syetems is the challenge we aim to address and the additive personalization is our proposed approach to address the challenge. So they are not seperated. FedRAP learns a user-specific item embedding and a globally shared item embedding jointly to enhance the performance in federated recommendation.
>
>
> ## W4 (Organizational Refinements)
>
> We will improve the organization and logical flow of the paper as you suggested in the updated draft.
>
> ## W5 & Q1 (Clarification on ‘Reconstruction Error’)
>
> In our paper, the term "reconstruction" refers to predicting the user ratings given in the training data. The reconstruction or prediction of ratings is achieved by the inner product between each user embedding $\mathbf{u}_i$ and each item embedding, which adds the global item embedding $\mathbf{C}$ and local item embedding $\mathbf{D}^{(i)}$ together. The reconstruction error compares the prediction with the ground truth ratings by Eq. (1).
>
> ## W6 (Questioning Novelty)
>
> FedRAP addressed a key challenge in federated recommendation and achieved significant improvement compared to existing methods on standard benchmarks. Its performance is unprecedently on par with the centralized recommendation system performance. This is due to the following technical novelties:
> (1) Additive modeling combining a local item embedding with a sparse global item embedding, which has not been studied for the problem;
> (2) A curriculum learning approach with two regularizations, which is critical to achieve the proposed additive modeling.
>
> ## W7&Q4 (Comparative Analysis)
>
> We will cite both PerFedRec and PerFedRec++ and discuss them in our uploaded draft. We are still tuning the hyperparameters for PerFedRec and aiming to include it in our experimental comparisons. Since PerFedRec++ has not released its code and it is a very recent preprint (published later than our preprint), we will try our best to include it in the future version of our paper.

---

> > ### Author Response · Authors · 2023-11-13
> > **Rebuttal for Q3 (Sparse Embedding) by Authors**
> >
> > ## Q3 (Sparse Embedding)
> >
> > Table R2. Sparsity of the global embedding matrix $\mathbf{C}$ learned by FedRAP-L2 and FedRAP on the ML-100K dataset. **Difference** represents the percentage of different elements (whose differences are above a threshold) between $\mathbf{C}$ learned by FedRAP and FedRAP-L2.
> >
> > |             |  >1e-2 |  >1e-3 |  >1e-4 |  >1e-5 |  >1e-6  |
> > |-------------|:------:|:------:|:------:|:------:|:-------:|
> > | FedRAP-L2   | 79.39% | 98.05% | 99.83% | 99.99% | 100.00% |
> > | FedRAP      | 52.68% | 74.55% | 93.46% | 98.65% |  99.84% |
> > | Difference  | 26.71% | 23.50% |  6.37% |  1.34% |  0.16%  |
> >
> > We explored the impact of the sparsity regularization on $\mathbf{C}$ in FedRAP by ablation study in Section 4.5, where we set the feature dimension of the item embedding to 16. We first evaluate the sparsity by calculating the percentage of elements in $\mathbf{C}$ exceeding a specified threshold for both FedRAP-L2 and FedRAP after the 100-th training iteration, as presented in Table R2. It shows that as the threshold increases, the difference between the two methods on sparsity gradually grows. Specifically, when the threshold is set to 1e-2, FedRAP, compared to FedRAP-L2, achieves a 33% reduction in communication overhead.
> >
> > Furthermore, we randomly selected 16 items from the item set and visualized the corresponding distribution of entries in $\mathbf{C}$ after the 100th training iteration. As shown in the first column of Figure 7, $\mathbf{C}$ learned by FedRAP is sparser than $\mathbf{C}$ learned by FedRAP-L2.
> >
> > These findings support our hypothesis that the user preferences affects item ratings and personalized item embedding can better capture the data heterogeneity. Moreover, they validate FedRAP's capability to learn better personalized models for each user.

---

> > ### Comment · Reviewer_QwLj · 2023-11-23
> > **Response to the authors' rebuttal**
> >
> > I appreciate the authors' rebuttal. My major concerns on the motivation, novelty, and experiments have been addressed in the rebuttal.

---

> ### Author Response · Authors · 2023-11-16
> **Rebuttal for W7&Q4 (Comparative Analysis) by Authors**
>
> ## W7&Q4 (Comparative Analysis)
>
> We conducted a comparative analysis of PerFedRec and FedRAP on ML-100K and ML-1M. The results, as shown in Table R3, indicate that FedRAP demonstrates competitive performance compared to PerFedRec. We are still tuning the hyperparameters for PerFedRec on more datasets.
>
> Table R3. Experimental results of PerFedRec and FedRAP shown in percentages on the ML-100K and ML-1M dataset.
> |           | ML-100K | ML-100K |  ML-1M |  ML-1M  |
> |-----------|:-------:|:-------:|:------:|:-------:|
> |           |  HR@10  | NDCG@10 |  HR@10 | NDCG@10 |
> | PerFedRec |  63.56  |  44.40  | 63.84  |  43.99  |
> | FedRAP    |  97.09  |  87.81  | 93.24  |  71.87  |

---

> ### Author Response · Authors · 2023-11-21
> **[Discussion ends in <2 days] Would you mind confirming if your concerns have been addressed?**
>
> Dear Reviewer QwLj,
>
> In the rebuttal, we have rephrased the motivation and novelty, highlighting the role of sparse constraints on $\mathbf{C}$. We have conducted a comparative analysis of PerFedRec and FedRAP on ML-100K and ML-1M, with the results presented in Table R3. Furthermore, we are refining the manuscript to enhance consistency in expression and reduce typographical errors. In future releases, we plan to cite and discuss PerFedRec++ extensively.
>
> As we are approaching the midpoint of the discussion period, we would like to cordially inquire about the extent to which we have successfully addressed the concerns outlined in your review. Should there be any lingering points that require further attention, please rest assured that we are enthusiastic about the opportunity to provide comprehensive responses to any subsequent queries or comments you may have.
>
> Your constructive input remains invaluable to us, and we appreciate your dedication to enhancing the quality of our manuscript. Thank you for your time and consideration.
>
> Best,
>
> Authors

---

### Official Review · Reviewer_MAPK · 2023-10-31

**Soundness:** 3 good
**Presentation:** 3 good
**Contribution:** 3 good
**Rating:** 8
**Confidence:** 4

**Summary:**

Federated Recommendation Systems (FRS) is a promising field, however, existing methods not pay enough attention to personalization . To enhance personalization in the field of FRS, FedRAP proposed by the authors address the challenges of heterogeneity and personalization in FRSs. FedRAP decouples the common knowledge and personalized knowledge by decoupling user embedding to on-server (updated by server-client communication) sparse one and on-client one. Regularization is used to restrict training.

**Strengths:**

1. It points out the shortcomings of the existing FRS methods, and gives a complete solution, which is instructive.
2. The framework is complete, and the paper is clearly expressed.
3. Theoretical analysis is sufficient. Proofs of error and differential privacy in federated learning are comprehensive.
4. Limitations about FedRAP are detailed

**Weaknesses:**

None

**Questions:**

None

---

> ### Author Response · Authors · 2023-11-13
> **Thank you for your constructive comments and support!**
>
> We would like to appreciate you for your constructive comments and support!

---

### Official Review · Reviewer_eLAA · 2023-11-01

**Soundness:** 3 good
**Presentation:** 3 good
**Contribution:** 3 good
**Rating:** 8
**Confidence:** 5

**Summary:**

This paper studies the problem of federated recommendation and proposes a new method improving the personalization performance. The authors developed a new additive item embedding model that combines a sparse global embedding with a user-personalized embedding, where the federated recommendation only needs to periodically send the sparse embedding between servers and users, and thus saves the communication cost. The user-personalized embedding is only stored at the user devices and optimized for each user so the personalization performance can be improved. They developed a curriculum to facilitate the training of the two embeddings, which learns the personalized ones at first before adding the sparse global one. This method achieved significant improvements on five benchmark datasets of different domains and sizes. The paper also provides a differential privacy analysis to the proposed approach.

**Strengths:**

1. Federated recommendation is an important problem that has not been fully explored in the federated learning community. Improving the personalization performance is an open key challenge in the federated recommendation setting.
2. The proposed additive item embedding model is a novel contribution to the field and addresses two critical challenges in the federated recommendation area, i.e,. reducing communication cost and personalization with knowledge sharing.
3. The ablation study shows that the curriculum is important to learn the item embedding model.
4. The performance achieved by the proposed FedRAP is remarkable in Figure 1: on six widely-used benchmarks, it outperforms all the federated recommendation baselines by a substantial margin and largely reduces the gap between federated algorithms and centralized ones---this is nontrivial and worth being highlighted.
5. The ablation study includes 8 variants of the proposed methods and demonstrates the advantage of most components in the proposed FedRAP. The analysis and experiments regarding differential privacy make their method more trustworthy.
6. They provided the code, which facilitates reproducible research in the future. I believe this work with the code released can be very interesting to people working in the area of federated recommendation systems.

**Weaknesses:**

1. The variants in the ablation study introduced in Section 4.5 do not come with intuitive explanations of the motivations for comparisions with them. They are unclear to readers if they do not carefully read the following analysis.
2. There are some typos and grammar errors which should be corrected.

**Questions:**

1. Can you provide an ablation study of the regularization in Eq2?
2. "FedRAP-No: Removes item diversity constraint on C" --- should "diversity" be "sparsity"? Is this a typo?
3. In Figure 7, it seems that each user-personalized matrix is row-sparse with most rows to be all-zeros. How do we understand this row-sparsity? Can you (or did you) take advantage of this row-sparsity to further reduce the communication cost (e.g., by only sending the non-allzero rows)?
4. How do we justify whether the degradation numbers in Table 1 are large or small? Are there any baselines' degradation results to be compared with?

Overall, I think this is a solid work on an important open problem. Their proposed item embedding model has a novel design successfully addressing the communication and personalization problems in federated recommendation. Their reported improvement (when compared to existing federated recommendation approaches) on six well-known benchmark datasets is impressive and significant, demonstrating the effectiveness. The ablation study and differential privacy analysis are thorough and further strengthened the work. The paper is clear overall but there is still some room for improving the writing/presentation. Some questions above need further clarifications.

---

> ### Author Response · Authors · 2023-11-13
> **Rebuttal by Authors**
>
> Thank you for your constructive comments and support!
>
> ## W1 (Unclear Motivations)
> In Section 4.5, we introduced eight variants of FedRAP based on three ablation study tasks. Specifically:
> 1. To investigate the roles of global item embedding and local item embedding in FedRAP and validate the basic assumption of user preference influencing ratings, we designed FedRAP-C using only global item embedding and FedRAP-D using only local item embedding.
> 2. To explore the impact of different constraints on C, we introduced FedRAP-No, which imposes no constraints on C, and FedRAP-L2, which uses the Frobenius norm on C.
> 3. To examine the influence of different weight curricula on FedRAP's performance, we introduced FedRAP-fixed using two constants $v_1$ or $v_2$ in hyperparameters, FedRAP-sin using a sine function for hyperparameter variation, and FedRAP-frac using fractional changes in hyperparameters. Additionally, we presented FedRAP-square with a hyperparameter periodically transitioning between zero and constants $v_1$ or $v_2$.
>
> These three ablation study aim to validate the contributions of three key innovations in FedRAP: two-way personalization, encouraging the sparsity of C, and a curriculum transitioning from full to additive personalization.
>
> ## W2 (Improve Writing)
> Thank you for bringing this to our attention! We will thoroughly review the paper and address any typos and grammar errors to ensure its clarity and quality. Your feedback is valuable in enhancing the overall presentation of the work.
>
> ## Q1 (Ablation Study)
>
> We introduce a new variant of FedRAP called FedRAP-S, where we remove the regularization $- \lambda_{(a, v_1)} ||\mathbf{D}^{(i)} -  \mathbf{C}||^2_F$ encouraging the difference between the global item embedding $\mathbf{C}$ and local item embedding $\mathbf{D}^{(i)}$ from Eq. (3). We trained FedRAP-S five times on the ML-100K dataset and report the results in Table R1, where FedRAP-S shows a significant performance degradation compared to FedRAP. This degradation could be attributed to the absence of constraints between user-specific item information and globally shared item information in FedRAP-S. Without such constraints, the personalized model becomes overly individualized, leading to overfitting on user data and a decrease in generalization ability. Therefore, enforcing differentiation serves as a bridge between personalized and shared models and controls the complexity of the personalized component and preventing overfitting. This ensures a balance between personalization and generalization in the model.
>
> Table R1. Experimental results of FedRAP and FedRAP-S shown in percentages on the ML-100K dataset.
> |          |      HR@10     |     NDCG@10    |
> |----------|:--------------:|:--------------:|
> | FedRAP   | 97.09$\pm$0.56 | 87.81$\pm$2.00 |
> | FedRAP-S | 65.86$\pm$1.34 | 39.99$\pm$0.46 |
>
>
> ## Q2 (Typo)
>
> It should be "FedRAP-No: Removes item sparsity constraint on C" instead of "diversity." This is indeed a typo. We appreciate your keen observation, and assure you that this error will be corrected in the future version of the paper.
>
> ## Q3 (Explanation of Fig. 7)
>
> In Fig. 7, the row-sparsity pattern in each user-personalized matrix, denoted as local item embedding  $\mathbf{D}^{(i)}$, indicates non-zero elements corresponding to items the i-th user has interacted with. Conversely, all-zero rows signify items that the user has not accessed. To address this, global item embedding $\mathbf{C}$ is introduced to incorporate global item information, allowing FedRAP to predict whether a user will interact with items they have not engaged with before. It's noted that optimizing the storage of $\mathbf{D}^{(i)}$ can reduce client-side space cost. However, since FedRAP transmits only $\mathbf{C}$ between the client and server, optimizing the storage of $\mathbf{D}^{(i)}$ does not lead to a reduction in communication costs.
>
>
> ## Q4 (Baseline Comparison)
>
> We have evaluated the performances on the Movielens-100K (ML-100K) dataset, comparing FedMF, FedNCF, PFedRec, and FedRAP with and without Differential Privacy (DP). Due to space constraints in the main text, the detailed results of our experiments are provided in Appendix Sec. B.4, as illustrated in Tables 5 and 6. The results demonstrate that FedRAP maintains a performance advantage, even in the presence of privacy-preserving measures. These findings showcase the robustness and efficacy of FedRAP in privacy-preserving federated recommendation scenarios.

---

### Meta-Review · Area_Chair_pAiJ · 2023-12-11

**Metareview:**

This paper proposes to address an important problem of federated learning for recommendation systems. The paper designs the embedding to with two parts, one is the sparse view on the server with dense view on the clients.

Strength:
1. The additive item embedding model in FedRAP appears to be novel and addresses key challenges in federated recommendation.
2. This paper presents comprehensive ablation study as well as good theoretical analysis.

Weakness:
1. The proposed method appears to be only applicable to linear model setting, which could limit the impact the approach.
2. Storing item embeddings can quickly become very large when the number of items becomes larger with federated recommendation system. This question might be slightly out of scope of this paper, but this issue can be the key to prevent the idea of paper to be useful in practice.

**Justification For Why Not Higher Score:**

The practical impact of this paper is limited.

**Justification For Why Not Lower Score:**

The proposed approach looks novel, empirical results are solid and could provide a useful reference for further development in this area.

---

### Decision · Program_Chairs · 2024-01-16

Accept (poster)